# Understanding CNNs as a model of the inferior temporal cortex: using mediation analysis to unpack the contribution of perceptual and semantic features in random and trained networks

**Anna Truzzi & Rhodri Cusack**[*]
Trinity College Institute of Neuroscience
Trinity College Dublin
Dublin 2, Ireland
`truzzia@tcd.ie, cusackrh@tcd.ie`

## Abstract

Convolutional neural networks (CNNs) trained for visual recognition can predict activity in primate inferior temporal cortex (IT). It was generally accepted that this is because training leads the CNNs to develop tuning to visual features similar to those in the brain. However, recent evidence that untrained random-weight CNNs explain IT variance to a similar magnitude appears inconsistent with this view. Since IT contains rich representations of both perceptual and semantic features, here we propose a resolution to this conflict, that random and trained networks capture different aspects of IT activity. Specifically, we hypothesised that random networks capture perceptual aspects of IT, while trained networks capture semantic aspects but not perceptual ones. We evaluated a trained standard AlexNet and an untrained random network shown to correlate better with the brain, DeepCluster. The ability of the CNNs to predict IT activity patterns and the role played by perceptual and semantic features was evaluated using regression models, multidimensional scaling, and mediation analysis. The results support the hypothesis and highlight that, whether CNNs are used as models of the brain, or the brain is used to inspire advances in neural networks, it is not enough to know how similar a given model is to the brain: we also need to know why.

## 1   Introduction

CNNs trained for object recognition - often AlexNet trained on ImageNet [1, 2, 3] - have proven to be good models of IT activity measured with neuroimaging or electrophysiology. It has been proposed that the discriminative features learned by CNNs during visual task training are similar to the features learned or evolved by the brain to solve vision based problems [4, 5]. However, there is increasing evidence that random-weight networks (i.e., networks without training) correlate with IT activity in human and animal models almost as well as trained CNNs, across analysis methods including representational similarity analysis and encoding models [6, 7, 8, 9]. In at least one instance, a network with random weights correlated with the brain even better than a trained CNN [10]. Specifically, Truzzi and Cusack [10] used the standard AlexNet architecture and an unsupervised network called DeepCluster, and found that the random DeepCluster correlated with IT significantly better then the trained DeepCluster or the trained/untrained AlexNet. In the case of DeepCluster, the training seemed to lead the network's representation away from IT's, rather than bringing it closer. It was found that this was not due to the type of training, since using a supervised protocol to train

---

[*]www.cusacklab.org

2nd Workshop on Shared Visual Representations in Human and Machine Intelligence (SVRHM), NeurIPS 2020.

the same DeepCluster architecture led to the same result. If the training process is not guaranteed to bring the CNNs representations closer to the brain's, we need to ask which features of IT are captured by random and trained CNNs. Human neuroimaging and electrophysiology has shown that IT represents both semantic information such as visual class or animacy [11, 12, 1, 13, 14] and perceptual information like spatial position [15]. Here we investigated the possibility that although random and trained CNNs correlate with IT with similar magnitude, they do so for different reasons. Our overarching hypothesis, was that random CNNs are similar to IT because they capture its perceptual aspects, while trained CNNs capture IT's semantic aspects but have lost the ability to capture its perceptual aspects. Indeed, for a CNN to solve the specific task of object recognition, the presence of irrelevant perceptual information may be a nuisance, and so this will be lost in training. In contrast, IT serves many downstream tasks of which object classification is just one, and perceptual features may be retained for these other tasks. We test the overarching hypothesis in three steps. First, we tested if a random network captures different aspects of IT representation compared to trained CNNs using linear models. Second, once we established that the two types of network captured different aspects of IT variability, we qualitatively investigated the nature of the representations in random and trained CNNs by visualizing them using multidimensional scaling. Third, to quantify which features, perceptual or semantic, drove the representational similarity between CNNs and IT, we used a mediation analysis.

## 2 Methods

### 2.1 Networks, neuroimaging, and preliminary analysis

**Standard AlexNet.** We used the standard AlexNet architecture [16] from torchvision.models, as used in previous studies [1, 2, 3]. The model was trained on ImageNet (top 1-accuracy: 56.6) and was not fine-tuned to the 92 visual stimuli used in the present study. We recorded activity at the output of the ReLUs in the five convolutional layers and the two fully connected layers.

**DeepCluster.** As in Truzzi and Cusack [10], we evaluated DeepCluster [17], trained in an unsupervised way on ImageNet (top 1-accuracy from 5th convolutional layer: 36.1) and again not fine-tuned to the 92 visual stimuli. The underlying convolutional network of DeepCluster can be an AlexNet [16] or a VGG16 architecture, both modified for unsupervised learning [17] with the local response normalisation layers removed and batch normalisation used instead [18] and an initial linear transformation based on Sobel filters applied on the input to remove colour and increase local contrast. Truzzi and Cusack [10] used the AlexNet instantiation but the results generalise to VGG16 (see Figure S1 in the appendix). To present a proof-of-principle we focus on the simpler AlexNet, and record activity at the output to the ReLU of the second convolutional layer of random DeepCluster as this was previously found to best correlate with IT [10].

**Neuroimaging and Representational Similarity Analysis.** Brain activity in response to 92 images was measured in the brains of 15 adults using functional MRI by Cichy et al. [19]. We compared how IT cortex and the CNNs represent these stimuli, using the well-established method of representational similarity analysis [20, 1], which calculates how dissimilar the brain activity evoked by each stimulus is to that evoked by every other stimulus (using *1-Pearson correlation*), yielding a representational dissimilarity matrix (RDM). For each layer of interest in the CNNs, an RDM was also calculated using a the same similarity analysis procedure. The IT and CNNs RDMs were then used in further analysis.

### 2.2 Do random and trained CNNs explain different aspects of IT variability?

If random and trained CNNs explain different aspects of IT, then together they should explain more variance than the trained CNN alone. To test this, we set up an ordinary least squares linear regression model at the level of the RDMs, which modelled the IT RDM as a weighted combination of one layer of the trained (AlexNet layer 7) and one layer of the random (DeepCluster layer 2) CNNs. These two layers were chosen because we expected their internal representations to be driven by mostly perceptual or semantic features, respectively. We used a method from a recent preprint Storrs et al. [8] that cross-validates bootstrapping across visual classes as well as across subjects (see Appendix for more details). However, any predictive power obtained in this model could be simply the result of the combination of earlier and upper layers of the networks. Layer 2 of random DeepCluster could be explaining no more IT variability than early layers of trained standard AlexNet, especially when

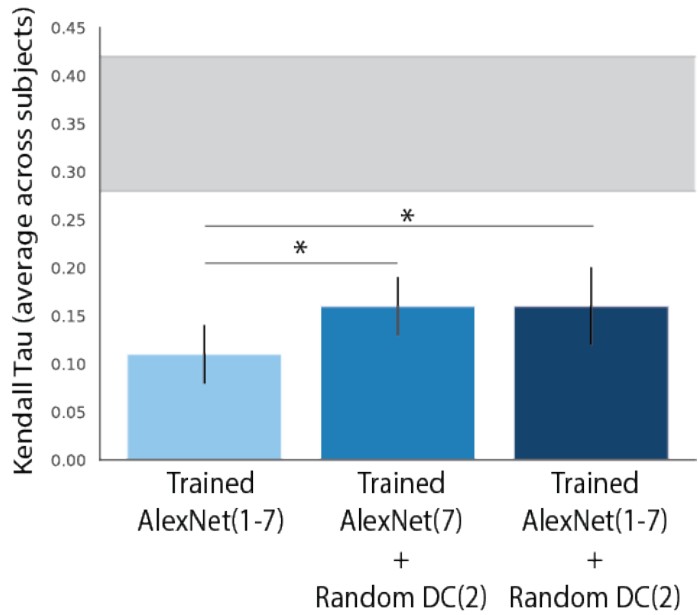

Figure 1: Models including random DeepCluster - DC in the figure - (dark and mid blue) explained IT representation better than models including trained AlexNet alone (light blue). Bars show the mean and standard deviation of correlation between the representations predicted by the models and the true IT representations across folds and bootstrapping runs. The gray stripe shows the noise ceiling, the maximum correlation that could be expected given the noise in the IT measurements.

considering that layer 7 of trained standard AlexNet did not have the strongest correlation with IT. To account for this possibility, two more models were run. The second model modelled IT from a weighted combination of all layers of trained standard AlexNet. The third model was the same as the second, except that it also included layer 2 of random DeepCluster. We only tested a single layer from random DeepCluster because our goal was to create the most parsimonious model in which to test if representations from random DeepCluster will improve modelling of IT. Finally, trained standard AlexNet and random DeepCluster differ in both training and the architecture. To isolate the effect of training, we ran two final models with random standard AlexNet: i) one model including all layers of random standard AlexNet, ii) a second model combining all layers from trained standard AlexNet and layer 2 from random standard AlexNet. These regression models were tested against chance and contrasted by calculating bootstrapped confidence intervals (see Appendix for more details).

### 2.3 Contribution of semantic vs perceptual features.

**Multidimensional scaling (MDS).** To qualitatively explore how images were represented in layer 2 of random DeepCluster and layer 7 of trained standard Alexnet we visually inspected their MDS plots. MDS was chosen over tSNE as it better preserves the distance structure at the global level.

**Mediation analysis.** To quantitatively test how semantic and perceptual characteristics mediate the relation between the CNNs and IT, we used mediation models (see Appendix for more details). For each image in the stimulus set, two semantic (animacy, semantic category) and seven perceptual (size, thinness, orientation, silhouette, contrast, hue, luridness) features were assessed (see Appendix for more details) and each feature's RDM was calculated. One mediation model was run for each CNN, with the IT RDM averaged across subjects as dependent variable, the layer's RDMs (layer 2 of random DeepCluster vs layer 7 of trained standard AlexNet) as the independent variable, and the feature RDMs as multiple mediators. From the linear modelling we found that combining these two layers, layer 7 of trained standard AlexNet + layer 2 of random DeepCluster, was able to explain IT variability better than a model including all layers from trained standard AlexNet and as well as a model combining all layers of trained AlexNet with layer 2 of random DeepCluster. Applying the mediation model to each of these two layers separately allowed us to investigate why the combination

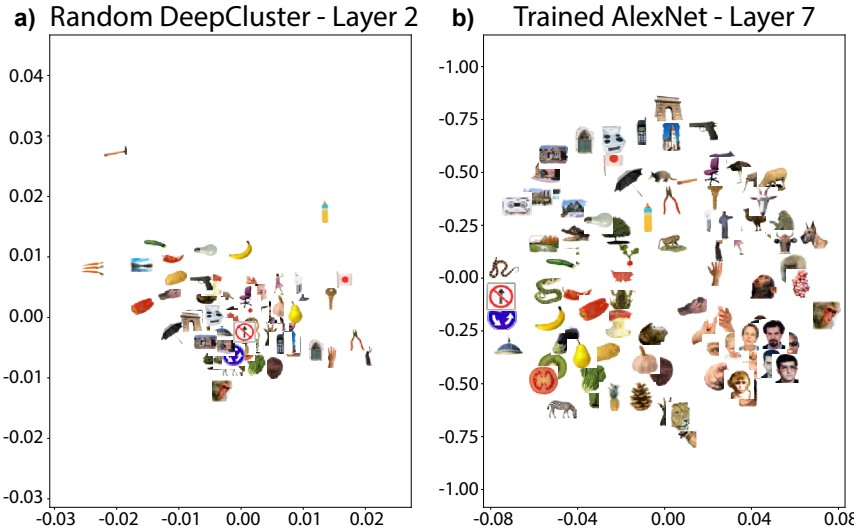

**a)** Random DeepCluster - Layer 2    **b)**    Trained AlexNet - Layer 7

Figure 2: Multidimensional scaling plots showing whether images from the 92 stimuli set are close or far apart in the representational space. a) Images as represented by layer 2 of random DeepCluster. b) Images as represented by layer 7 of trained standard AlexNet

of layer 7 of trained standard AlexNet and layer 2 of random DeepCluster predicted IT over and above the combination of all the trained standard AlexNet layers. For each indirect path we calculated its proportion of the total regression weights (i.e., indirect path+direct path). Then, we calculated two aggregate sums for the perceptual, and for the semantic features. For the mediation models, cross-validation was not necessary because our goal was not to predict IT activity but to understand what drives the prediction.

## 3    Results

### 3.1    Do random and trained CNNs explain different aspects of IT variability?

All three main models significantly correlated with IT *(trained AlexNet(1-7) Kendall's $\tau$ CI = [0.04,0.18]; trained AlexNet(7)+random DeepCluster(2) CI = [0.10,0.23]; trained AlexNet(1-7)+random DeepCluster(2) CI = [0.10,0.23])*. Critically, the two models containing random network *(trained AlexNet(7)+random DeepCluster(2); AlexNet(1-7)+random DeepCluster(2))* correlated more strongly with IT than the trained standard AlexNet *(differences in Kendall's $\tau$, respectively, CI = [-0.12,-0.5e-04] and CI = [-0.12,-0.01])*. The two models containing the random network did not differ from each other in their prediction of IT *(CI = [-0.03,0.03])*. In summary, the RDM of the random CNN explained variance in the IT RDM, over-and-above the RDM of the trained CNN, supporting our hypothesis that they carry independent predictive value. On the other hand, the model including all layers from random standard AlexNet did not significantly predict IT *(random AlexNet(1-7) Kendall's $\tau$ CI = [-0.37,0.08]*. The model combining all layers of trained standard AlexNet with layer 2 of random standard AlexNet, although significantly different from baseline *(trained AlexNet(1-7)+random AlexNet(2) Kendall's $\tau$ CI = [0.04,0.18]*, did not significantly differ from the model including all layers from trained AlexNet only *(differences in Kendall's $\tau$ CI = [-0.01,0.02]*. Instead, both models including layer 2 of random DeepCluster, that is trained AlexNet(7)+random DeepCluster(2) and trained AlexNet(1-7)+random DeepCluster(2), predicted IT better than the model including layer 2 or random standard AlexNet *(differences in Kendall's $\tau$, respectively, CI = [0.0002,0.12] and CI = [0.007,0.12]*. Therefore, the representations in random standard AlexNet do not capture variability in IT and do not contribute to explain IT over and above the representations in trained standard AlexNet. Random DeepCluster is therefore able to capture some variability in IT that both trained and random standard AlexNet are missing. This difference is likely driven by the presence in DeepCluster of an initial Sobel filter that increases local contrast, or

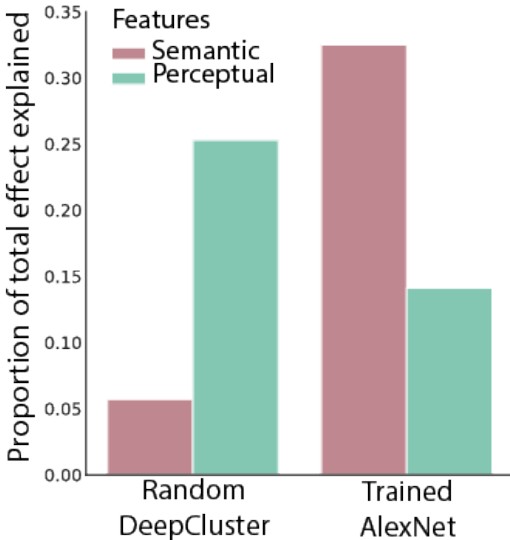

Figure 3: The proportion of IT's RDM explained by random DeepCluster and trained AlexNet that was mediated by the semantic and perceptual features.

of the batch normalization. Indeed, when adding the same modifications to the VGG16 architecture we saw that random modified VGG16 correlated with IT better than the trained modified VGG16 (see Figure S1 in the appendix). It should be noted that none of the models captured all the explainable variability in IT. However, the magnitude of the variance explained by these models is in line with other results in the literature [21].

### 3.2   Visualizing representations of random and trained CNNs.

Multidimensional scaling (MDS) places images that evoke more similar patterns of activation closer together. The MDS plots showed that in layer 2 of random DeepCluster the stimuli seemed grouped by shape, with horizontal thin objects towards the upper left and vertical thin objects to the lower right. In contrast, in layer 7 of trained standard AlexNet the stimuli were grouped by category [Fig.2]. Interestingly, the shape-based grouping seen in layer 2 of random DeepCluster was not present in layer 2 from random standard AlexNet, where objects seemed to rather be grouped based on colour (see Fig.S2 in the appendix).

### 3.3   Mediation analysis of contribution of semantic vs perceptual features.

For random DeepCluster perceptual features explained a higher proportion of the total variance than the semantic features. In contrast, for trained standard AlexNet the semantic features combined explained a higher proportion of the total effect compared to the perceptual features [Fig.3]. A break down of the proportion for the indirect path of each feature is shown in Table 1. The full model results for both layer 2 random DeepCluster and layer 7 trained standard AlexNet are reported in Tables S1 and S2, respectively, in the appendix.

## 4   Discussion

Even when the representations of random and trained networks correlated with IT to a similar degree, they explained different aspects of its representations. Adding only one layer of a random CNN to the model predicting IT activity was already enough to double the explained variance. This finding highlights that a random DeepCluster captures aspects of IT representation that the trained standard AlexNet misses. IT has rich representations, likely to sustain many downstream tasks in addition to object recognition. In contrast, CNNs are typically trained for just one task. The training process introduces tuning for discriminative features while leading to a reduction in the coding of

Table 1: The strength of mediation by each feature was quantified by its indirect effect divided by the sum of indirect and direct effects for the model.

| Type | Mediating feature | Random DeepCluster | Trained AlexNet |
|---|---|---|---|
| Semantic | category | -0.1% | 2.6% |
| | animacy | 5.8% | 30.0% |
| Perceptual | size | -0.6% | -4.2% |
| | thinness | 10.6% | 0.8% |
| | orientation | 0.1% | 0.9% |
| | silhouette | 14.2% | 14.6% |
| | contrast | 0.5% | -0.1% |
| | hue | -1.0% | 0.3% |
| | luridness | 1.5% | 1.8% |

features that could be a nuisance for that particular task. However, the representations that are a nuisance for one task will be valuable for another. Even in the context of object categorization, IT may be "cheating" and using shortcuts based on simple perceptual cues, rather than full invariant representations [22, 23, 24]. CNNs use this strategy as well, but previous findings have shown that for object recognition humans prioritize global features, such as the shape of objects, while CNNs prioritize local features, such as their texture [25]. This may cause the CNNs to learn, during training, to disregard some mid-level perceptual features that are instead retained in IT. If random CNNs are sensitive to shape information, this could explain some of the variance it captures in IT. Supporting this, the mediation analysis confirmed that at least one shape measure, thinness, mediates the relation between random DeepCluster and IT, whereas it does not mediate the correlation between the trained network and IT. A limitation of the current study is that only two networks were considered and extension to a wider range of architectures, as well as the use of different random seeds would further the understanding of what makes CNNs representations similar to neural activity patterns in IT and of the possibility to generalise the present findings. Nonetheless, our study already highlights that **knowing how well CNNs and IT correlate is not enough, we also need to understand why** [26]. Understanding what features make the internal representations of CNNs similar to neural activity patterns in human IT will impact both neuroscience and AI [23]. In neuroscience, knowing what type of information is processed in the ventral visual stream will give us insights on what the brain uses to learn about the world. On the other hand, building CNNs' architectures that uses the same features to learn about their visual input could improve the networks' performance and make it more generalizable and robust to adversarial attacks.

## Broader Impact

Outputs from this research will advance the knowledge in neuroscience and AI benefiting researchers in those fields. Findings from the present project will not put anyone at a disadvantage. No new system is tested in the current research.

## Acknowledgements

Funded by the ERC Advanced Grant ERC-2017-ADG, FOUNDCOG, 787981 We thank Radoslaw Martin Cichy for making the fMRI data available through the Algonauts project. Authors declare no conflict of interest.

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

# Appendix

**Cross-validation across subjects and visual classes.** Like [8] we used bootstrapping to calculate a noise ceiling measure and variability estimate. For each model, 20000 bootstraps were run with 20 cross-validation folds each. Within each fold, 12 image classes and 4 subjects were assigned to the test set. The bootstrap procedure was run with replacement, therefore more then one occurrence of the same subject and image class could be included in the same test set. However, the train and test set were mutually exclusive: the same subject or the same image classes were not present in both the training and test sets of a single fold. A linear regression model was used to fit the portion of the CNN layer RDMs to the IT RDM using only the training image classes. The IT RDM was the average of the training subjects' IT RDMs. This model was then used to predict the representational similarity of the test image classes, by using the fitted coefficients to re-weight the test portion of the layer RDMs and summing them. This was then compared to the IT RDMs for the test image classes for each individual test subject's RDM, using Kendall's $\tau$ as the correlation statistic. The use of Kendall's $\tau$ is recommended because, being a rank correlation coefficient, it does not assume the presence of a linear relationship between the RDM values [1], and because it proved to be more likely than Spearman's Rho to prefer the true model over a simplified categorical model [27]. The final correlation score for each of the two models was calculated by averaging the Kendall's $\tau$ values across cross-validation folds. Within the bootstrap and cross-validation folds, the lower and upper bounds of the noise ceiling were also calculated. The calculation of the noise ceiling shows how much of the data variability is explained by our model by taking into account the noise intrinsic in the MRI data. If the correlation between the model and the MRI data is close to the noise ceiling, the model explains the data well [1]. The lower bound was the average correlation of the mean of the IT response from the training subjects to each individual test subject. The upper bound was the average correlation of the mean of the IT response across all subjects, both test and training, to each individual test subject. An aggregate summary of the noise ceiling was then obtained by averaging the lower and upper values across folds.

**Bootstrapping confidence interval.** The significance of each model against the baseline was calculated using the percentile bootstrap procedure with alpha = 0.05. The correlation values obtained across bootstrapping were sorted and the 95% confidence interval was calculated. To compare the models, the confidence interval was calculated on the difference between models. First the correlation values obtained across bootstrapping for one model were subtracted from the ones obtained for a second model. Then, the values obtained after the subtraction were sorted and the 95% confidence interval was calculated.

**Mediation models** A mediation model calculates direct and indirect relation between independent and dependent variables. The direct path calculates the relation between independent and dependent variables when the mediation factors remain unaltered, whereas the indirect path calculates how strongly other factors mediate the relation between the independent and dependent variables [Fig.**??**].

**Images' semantic features.** The semantic category was defined using the ones proposed in Khaligh-Razavi and Kriegeskorte [1]: human body, human face, non-human body, non-human face, inanimate natural, inanimate artificial. The RDM of the semantic categories was computed by assigning zero to each pair of images if they belonged to the same category and one if they belonged to different categories. In a similar way, the animacy of each image was defined and the RDM of animacy categories was computed by assigning zero to each pair of images if they were both animate or both inanimate, and one if they belonged to different categories.

**Images' perceptual features.** Seven perceptual features were calculated following Cusack et al. [22]: size, thinness, orientation, silhouette, contrast, hue, luridness. Size was measured as number of non-background pixels in the image. Thinness was calculated as the ratio of the first two singular values of the Cartesian coordinates of the occupied pixels. Orientation was assessed by calculating how many radians off the horizontal axis the primary axis was (the arctan of the ratio of the two elements in the first row of the singular value decomposition). The silhouette RDM was defined as the correlation between the silhouettes of the pair of images. Contrast was calculated as the mean of sum of squared difference of each color channel from the background color, for non-background pixels. Color hue was defined as the H value of mean color in HSV space, again for non-background pixels. Color luridness was calculated as the mean of standard deviation across RGB for each pixel, for non-background pixels. To create an RDM for each perceptual feature, the Euclidean difference for the feature for each pair of images was computed.

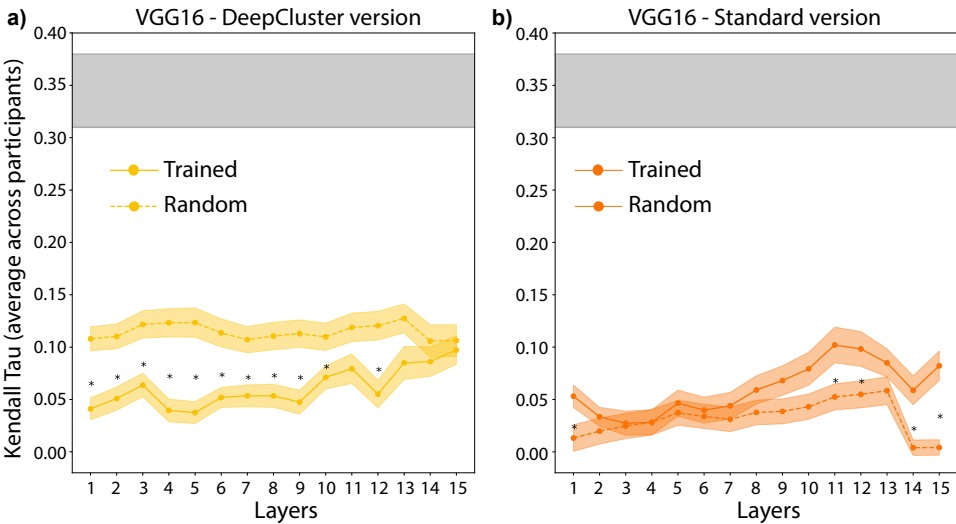

Figure S1. Results of the correlation between each subject's RDM and each networks' layer RDMs for two architectures: DeepCluster VGG16 modified for unsupervised learning and standard VGG16. The dashed lines represent the correlations with the random networks. The solid lines represents the correlations with the trained networks. The grey band represents the noise ceiling for the MRI data. Asterisks show in which layers one version of the network correlated with IT better than the other. a) Results for DeepCluster VGG16. b) Results for standard VGG16

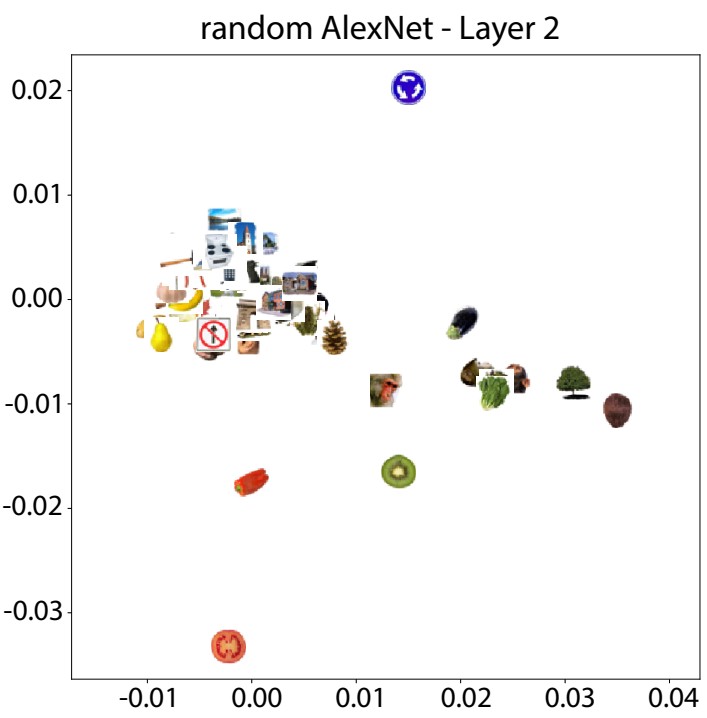

Figure S2. Multidimensional scaling plot showing the representational space of layer 2 of untrained random standard AlexNet.

Table S1. Mediation model testing whether the relation between layer 2 of random DeepCluster and IT is mediated by semantic or perceptual features.

| path | coef | se | p value | CI[2.5%] | CI[97.5%] | Sig |
|---|---|---|---|---|---|---|
| category $\sim$ X | -0.20 | 0.76 | 0.79 | -1.70 | 1.29 | No |
| animacy $\sim$ X | 3.69 | 1.00 | 0.0002 | 1.73 | 5.66 | Yes |
| size $\sim$ X | 1.29e+05 | 8.29+03 | 0.000 | 1.13e+05 | 1.45e+05 | Yes |
| thinness $\sim$ X | 3.58e+01 | 0.96 | 0.000 | 3.39e+01 | 3.77e+01 | Yes |
| orientation $\sim$ X | 1.68e+01 | 1.01 | 0.000 | 1.48e+01 | 1.88e+01 | Yes |
| silhouette $\sim$ X | -1.05e+01 | 3.91e-01 | 0.000 | -1.13e+01 | -9.77 | Yes |
| contrast $\sim$ X | 1.17e+05 | 3.52e+04 | 0.001 | 4.78e+04 | 1.86e+05 | Yes |
| hue $\sim$ X | 3.13e+02 | 6.38e+01 | 0.000 | -4.38e+02 | -1.88e+02 | Yes |
| luridness $\sim$ X | 2.21e+02 | 2.49e+01 | 0.000 | 1.72e+02 | 2.70e+02 | Yes |
| Y $\sim$ category | 0.01 | 0.003 | 0.000 | 0.009 | 0.02 | Yes |
| Y $\sim$ animacy | 0.06 | 0.002 | 0.000 | 0.05 | 0.006 | Yes |
| Y $\sim$ size | -5.59e-07 | 2.54e-07 | 0.03 | -1.06e-06 | -6.14e-08 | Yes |
| Y $\sim$ thinness | 0.03 | 0.002 | 0.000 | 0.02 | 0.03 | Yes |
| Y $\sim$ orientation | 0.004 | 0.002 | 0.09 | -5.58e-04 | 0.008 | No |
| Y $\sim$ silhouette | -0.07 | 0.006 | 0.000 | -0.08 | -0.06 | Yes |
| Y $\sim$ contrast | 2.12e-07 | 5.02e-08 | 0.000 | 1.14e-07 | 3.11e-07 | Yes |
| Y $\sim$ hue | 5.68e-05 | 2.78e-05 | 0.04 | 2.19e-06 | 1.11e-04 | Yes |
| Y $\sim$ luridness | 3.83e-04 | 7.08e-05 | 0.000 | 2.45e-04 | 5.22e-04 | Yes |
| Total | 3.52 | 0.13 | 0.000 | 3.26 | 3.77 | Yes |
| Direct | 2.43 | 0.14 | 0.000 | 2.16 | 2.70 | Yes |
| Indirect category | -0.003 | 0.01 | 0.8 | -0.03 | 0.02 | No |
| Indirect animacy | 0.20 | 0.057 | 0.000 | 0.10 | 0.31 | Yes |
| Indirect size | -0.02 | 0.03 | 0.5 | -0.08 | 0.03 | No |
| Indirect thinness | 0.4 | 0.07 | 0.000 | 0.2 | 0.5 | Yes |
| Indirect orientation | 0.003 | 0.04 | 0.9 | -0.06 | 0.07 | No |
| Indirect silhouette | 0.50 | 0.65 | 0.000 | 0.38 | 0.62 | Yes |
| Indirect contrast | 0.02 | 0.008 | 0.000 | 0.006 | 0.03 | Yes |
| Indirect hue | -0.03 | 0.01 | 0.004 | -0.056 | -0.02 | Yes |
| Indirect luridness | 0.05 | 0.02 | 0.000 | 0.02 | 0.09 | Yes |

Table S2. Mediation model testing whether the relation between layer 7 of trained standard AlexNet and IT is mediated by semantic or perceptual features.

| path | coef | se | p value | CI[2.5%] | CI[97.5%] | Sig |
|------|------|-----|---------|----------|-----------|-----|
| category ~ X | 0.99 | 0.04 | 0.000 | 0.91 | 1.08 | Yes |
| animacy ~ X | 1.13 | 0.057 | 0.000 | 1.01 | 1.25 | Yes |
| size ~ X | 6.67e+03 | 4.92e+02 | 0.000 | 5.69 | 7.62e+03 | Yes |
| thinness ~ X | 0.05e-02 | 0.07e-02 | 0.45 | -0.08e-02 | 0.18 | No |
| orientation ~ X | 1.06 | 0.06 | 0.000 | 0.94 | 1.17 | Yes |
| silhouette ~ X | -0.57 | 0.02 | 0.000 | -0.62 | -0.53 | Yes |
| contrast ~ X | -5.57 | 2.08 | 0.79 | -4.62e+03 | 3.51e+03 | No |
| hue ~ X | 2.85e+01 | 3.74 | 0.000 | 2.12 | 3.58e+01 | Yes |
| luridness ~ X | 1.30e+01 | 1.47 | 0.000 | 1.01e+01 | 1.59e+01 | Yes |
| Y ~ category | 0.01 | 0.003 | 0.000 | 0.01 | 0.02 | Yes |
| Y ~ animacy | 0.06 | 0.002 | 0.000 | 0.05 | 0.06 | Yes |
| Y ~ size | -5.59e-07 | 2.54e-07 | 0.03 | -1.06e-06 | -6.14e-08 | Yes |
| Y ~ thinness | 0.03 | 0.002 | 0.000 | 0.02 | 0.03 | Yes |
| Y ~ orientation | 0.004 | 0.002 | 0.09 | -5.58e-04 | 0.008 | No |
| Y ~ silhouette | -0.07 | 0.006 | 0.000 | -0.08 | -0.06 | Yes |
| Y ~ contrast | 2.12e-07 | 5.02e-08 | 0.000 | 1.14e-07 | 3.12e-07 | Yes |
| Y ~ hue | 5.68e-05 | 2.79e-05 | 0.04 | 2.19e-06 | 1.11e-04 | Yes |
| Y ~ luridness | 3.83e-04 | 7.08e-05 | 0.000 | 2.45e-04 | 5.22e-04 | Yes |
| Total | 0.20 | 0.008 | 0.000 | 0.18 | 0.21 | Yes |
| Direct | 0.11 | 0.008 | 0.000 | 0.09 | 0.1 | Yes |
| Indirect category | 0.005 | 0.003 | 0.08 | -7.04e-05 | 0.01 | No |
| Indirect animacy | 0.06 | 0.003 | 0.000 | 0.05 | 0.07 | Yes |
| Indirect size | -0.008 | 0.002 | 0.000 | -0.01 | -0.005 | Yes |
| Indirect thinness | 0.002 | 0.002 | 0.40 | -0.002 | 0.005 | No |
| Indirect orientation | 0.002 | 0.002 | 0.42 | -0.003 | 0.006 | No |
| Indirect silhouette | 0.03 | 0.004 | 0.000 | 0.02 | 0.04 | Yes |
| Indirect contrast | -1.13e-04 | 4.07e-04 | 0.78 | -9.22e-04 | 6.30e-04 | No |
| Indirect hue | 5.16e-04 | 7.38e-04 | 0.53 | -7.77e-04 | 0.002 | No |
| Indirect luridness | 0.004 | 0.001 | 0.000 | 0.002 | 0.006 | Yes |

