# OpenReview forum: "Understanding CNNs as a model of the inferior temporal cortex: using mediation analysis to unpack the contribution of perceptual and semantic features in random and trained networks"
_NeurIPS.cc/2020/Workshop/SVRHM — SVRHM@NeurIPS Poster_

### Official Review · AnonReviewer3 · 2020-10-28
**Uninformative analyses due to the lack of control comparisons**

**Rating:** 4
**Confidence:** 3

**Review:**

This study was motivated by the finding that a specific variant of AlexNet, as used in DeepCluster, was better able to explain variation in inferior temporal cortex (IT) during object recognition when it was untrained instead of being trained with ImageNet (Truzzi & Cusack, 2020). The same study also suggested that the explanatory power of this untrained architecture even went above and beyond of that of a standard AlexNet architecture trained on ImageNet. This surprising finding led the authors of the current paper to speculate that the explanatory power of this architecture is rooted in perceptual rather than semantic features. The central goal of this paper was thus to disentangle why these random weights explain variation in IT.

At the start of their paper, the authors make the argument that the features of both of these models (trained AlexNet and untrained DeepCluster Alexnet, random DC(2)) are complementary. To support their claim, they present an analysis in which they compare the fits to IT between the trained AlexNet features in isolation and variants in which the AlexNet features are accompanied by the random features of the other model (Random DC (2)). Unfortunately, the authors leave it unclear how much variance can be explained by the RandomDC (2) features alone in figure 1. Consulting the first study by Truzzi & Cusack (2020) suggests that RandomDC(2) can account for a similar correlation as reported here for the combined features (middle and right bar, figure 1). This makes the results in figure 1 less convincing, since it suggests that there appears to be shared variance between the trained AlexNet and the RandomDC(2) features. If both of these feature sets were complementary, one would expect a larger increase from the correlation reported by Truzzi & Cusack (2020) while both result in correlations of around 0.15.

While I appreciate the authors' reflection that models trained on ImageNet fail to capture the diversity of tasks that IT might be engaged in, I would have liked them to also consider the explanatory power of random parameters that add complexity to the model features that explain IT. In particular, is this result specific of this particular initialization? Can it reproduced across other random seeds?

Another crucial difference in the model features that I missed in the interpretation of the earlier findings is that the untrained AlexNet variant used in the first study used a preprocessing step in the form of Sobel filters. It would be valuable to isolate the effect of this addition to understand the explanatory power of this untrained architecture. This might be especially relevant since this most predictive layer, random DC(2), occurs so soon after this added computational step.

In their research question, the authors wonder whether these random DC(2) features were more perceptual rather than semantic and they answer this question by showing the features in MDS and with a mediation analysis. I liked that the authors adopted a range of visual characteristics and dissected their contribution with the mediation analysis as a general approach towards interpreting features of a deep neural network. I did however wonder whether the origin of these sets of features already foreshadows these results, with DC(2) features being much closer to the input of the network and AlexNet(7) much closer to the output layer? Could the same shift towards perceptual features be found if the second layer of the trained AlexNet was analysed with this approach? Adding this control analysis would clarify how this random DC architecture captures perceptual features beyond what can be expected due to its location in the architecture. What makes DC(2) specifically predictive?

In summary, while I agree with the authors about the relevance of their research question, this paper unfortunately does not clarify how this specific untrained AlexNet variant used in DeepCluster succeeds in explaining variability in IT.

---

### Official Review · AnonReviewer1 · 2020-10-30
**An Innovative Analysis of Neural Network Models of Human Visual Representation**

**Rating:** 7
**Confidence:** 4

**Review:**

This paper proposes an analysis for better understanding the predictive power of randomly initialized and trained neural networks as they relate to hemodynamic responses in human inferotemporal visual cortex. Proposing the main difference in these feature spaces to be the relative saturation of perceptual versus semantic information (the former supposedly superseding the latter as a function of training) , the authors use a combination of representational similarity, regression and mediation analysis to disambiguate the relative contributions of each kind of feature content to the overall prediction of neural activity.

The authors’ analysis is innovative and compelling, drawing on statistical methods often ignored in the larger machine learning literature. A few analytic choices are questionable (see below), but the main conclusions (that random and pretrained models explain different portions of neural variance and that the effects of the random models are mediated more by perceptual features than those of the pretrained models and vice versa) are overall supported by the results.

My main quibble is with the use of random DeepCluster AlexNet as opposed to random standard AlexNet as a point of comparison to standard pretrained AlexNet. Even if the authors have a theoretically motivated reason to believe that the architectural differences of random DeepCluster AlexNet contribute to a meaningfully different set of feature spaces than random standard AlexNet (which they don’t seem to have; see below), this choice commits a cardinal sin of empirical comparison, simultaneously modifying two variables (architecture and training) while attributing the results to only one (training). The authors suggest this choice doesn’t necessarily matter for the results, but if that’s the case, then the full control is preferable (modification of training alone).

This aside, I applaud the author’s efforts and look forward to seeing this work as it evolves further! Bravo.

Some questions (suggestions for clarification), in no particular order are listed below:
•	I assume layer 7 of standard pretrained AlexNet (like layer 2 of random DeepCluster AlexNet) was the layer most predictive of IT cortex? If not, this choice seems arbitrary, and possibly self-confirming if the theory that later layers of the networks contain more semantic information is true.
•	For analytic symmetry (and to be fully fair to random AlexNet, a model that contains layers 1-7 of random AlexNet should be included in the reported results).
•	If layer 7 of standard AlexNet explains as much variance as layers 1-7, what does this mean for the different quantity of perceptual and semantic features across the information processing hierarchy the authors propose in the outset?
•	The discrepancy between random standard AlexNet and random DeepCluster AlexNet is underexplored here. That the random network would cluster based on shape is very much incongruous with results that suggest convolutional networks represent images almost exclusively on the basis of texture (see Geirhos et al., 2019). Unless one of the architectural modifications (e.g. inclusion of batch normalizations) makes a significant difference in the representations one would only expect their representations to diverge by chance alone… This suggests results from random AlexNet may be somewhat unstable and somewhat suspect. What initialization was used? What are the details of the architectural differences between the two?
•	What regression was used in section 2.2? In ordinary least squares regression, an RDM can be assigned a negative coefficient, which is conceptually strange. Jozwik et al. (2016) use a nonnegative least squares regression to prevent this.
•	I assume the mediation analysis used the minimal version of the regression (one model RDM predicting one neural RDM); to better access the moderating effects of the different feature types across model layers, the authors might consider a moderated mediation, with model layer as the moderating variable.
•	The correlations here are generally low (maxing at around 0.15); how do simpler baselines perform? What is the noise ceiling? Contextualizing these values might alleviate concerns about their overall magnitude.

---

### Official Review · AnonReviewer2 · 2020-10-30
**Well written and interesting analysis**

**Rating:** 7
**Confidence:** 3

**Review:**

The authors ground their work on recent (yet to be established) claims of random untrained CNNs explaining IT variance just as well as pre-trained CNNs. In this paper, they propose a solution to those seemingly conflicting findings suggesting that pre-trained and random networks explain different aspects of IT: “semantic” and “perceptual” features respectively. This hypothesis was tested by 1) verifying that combined untrained+pretrained features explain more variance of IT than pretrained alone with linear regression, 2) a qualitative evaluation of the representations using “multidimensional scaling”, and 3) a quantitative evaluation using mediation analysis by extracting both semantic features (category, animacy), and perceptual features (size, thinner, orientation silhouette, contrast, hue, luridness)

Although the paper often points to supplementary material and figures that are not present in the document –– probably it is a shorter version of a longer manuscript –– it is well motivated and clearly written. Their take-home message of going beyond finding good CNN models and understand why they perform well is of paramount importance to the field.

Overall I have only minor comments/suggestions that could improve the work in the long run
* Although the three steps to test the hypothesis are sound. I found the second qualitative evaluation not very conclusive. Why was multidimensional scaling preferred over tSNE, or UMAP visualizations of the layer embeddings?
* I’m not too familiar with the paper that introduced DeepCluster (the random network that matched IT equally well), but I think it would be worth considering other architectures with random weights and explore how well they predict IT.
* Related to the last point, I would like to see consistency over several architectures for both pretrained and untrained networks to really isolate the effects of training.
* I found surprising that the category mediating feature contributed little variance (2%) compared to animacy (30%) for trained AlexNet. It would be interesting to investigate that further.
* What task would the authors propose to trained a network so that perceptual features are preserved and thus used for improving IT correlations?

---

### Public Comment · ~Anna_Truzzi1 · 2020-11-30
**Answer to reviewers**

We thank the reviewers for their valuable comments on our paper.
We revised the manuscript including the suggestions related to the current analysis and results, such as the addition of linear models testing the contribution of random standard AlexNet to the prediction of IT variability.
Moreover, we appreciate all the suggestions on how to move forward and, though they were not included in the current paper because out of its main scope, we aim to incorporate them in our next work which will address, between others, the use of different random seeds in the considered networks and of different networks.

---

### Decision · Program_Chairs · 2020-11-02

Accept (Poster)